# Recovery of Anthocyanins and Monosaccharides from Grape Marc Extract by Nanofiltration Membranes

**DOI:** 10.3390/molecules26072003

**Published:** 2021-04-01

**Authors:** Paul Muñoz, Karla Pérez, Alfredo Cassano, René Ruby-Figueroa

**Affiliations:** 1Department of Chemistry, Universidad Tecnológica Metropolitana, Las palmeras 3360, 7800003 Santiago, Chile; paul.munozm@utem.cl; 2Programa Institucional de Fomento a la Investigación, Desarrollo e Innovación (PIDi), Universidad Tecnológica Metropolitana, Ignacio Valdivieso 2409, 8940577 Santiago, Chile; karla.perezr@utem.cl; 3Institute on Membrane Technology, ITM-CNR, via P. Bucci, 17/C, I-87036 Rende, Italy

**Keywords:** wine by-products, grape marc, nanofiltration, anthocyanins, *Carménère*

## Abstract

Wastewaters and by-products generated in the winemaking process are important and inexpensive sources of value-added compounds that can be potentially reused for the development of new products of commercial interest (i.e., functional foods). This research was undertaken in order to evaluate the potential of nanofiltration (NF) membranes in the recovery of anthocyanins and monosaccharides from a clarified Carménère grape marc obtained through a combination of ultrasound-assisted extraction and microfiltration. Three different flat-sheet nanofiltration (NF) membranes, covering the range of molecular weight cut-off (MWCO) from 150 to 800 Da, were evaluated for their productivity as well as for their rejection towards anthocyanins (malvidin-3-*O*-glucoside, malvidin 3-(acetyl)-glucoside, and malvidin 3-(coumaroyl)-glucoside) and sugars (glucose and fructose) in selected operating conditions. The selected membranes showed differences in their performance in terms of permeate flux and rejection of target compounds. The NFX membrane, with the lowest MWCO (150–300 Da), showed a lower flux decay in comparison to the other investigated membranes. All the membranes showed rejection higher than 99.42% for the quantified anthocyanins. Regarding sugars rejection, the NFX membrane showed the highest rejection for glucose and fructose (100 and 92.60%, respectively), whereas the NFW membrane (MWCO 300–500 Da) was the one with the lowest rejection for these compounds (80.57 and 71.62%, respectively). As a general trend, the tested membranes did not show a preferential rejection of anthocyanins over sugars. Therefore, all tested membranes were suitable for concentration purposes.

## 1. Introduction

Winemaking activities generate huge amount of organic wastes which include by-products such as grape marc, lees, pomace, stalk and dewatered sludge, all of which require proper treatment before disposal [1]. Among them, grape marc contains high levels of biochemical oxygen demand (BOD) and chemical oxygen demand (COD); therefore, if not treated, it can be easily attacked by microorganisms generating foul odor and contamination problems inside the winery [2]. These environmental problems are a big issue for countries that produce wine.

The worldwide production of wine in 2018 was 292 million hl from 44.35 Mton of grapes. In South America, Chile is a recognized wine producer for both volume and quality of production. The Carménère variety is the emblematic grape of Chile and the production of red wine from this variety has reached about 70 million L/year [3]. Today, Chile is the sixth-largest wine producer in the world producing around 4.4% of the world’s wine. The country’s annual production grew from 0.2 million hectoliters in 1990 to 12.9 million hectoliters in 2018, climbing up the ranking of major world producers and exporters. [4,5]. It has been estimated that 33,000 tons of grape marc of Carménère are generated each year in Chile [3].

Grape marc, also known as grape bagasse or grape pomace, is the main solid residue generated in winemaking; it typically consists of skins, seeds, stalks and moisture that remain after pressing the grapes. The volume of these solids represent up to 20–30% of the total wine production [6]. Grape marc and lees represent a rich source of bioactive compounds such as polyphenols (2–6.5%) to which a wide range of biochemical activity is attributed [7]. From the chemical point of view, the polyphenols can be divided into flavonoids (flavonols, anthocyanins, and flavanols) and non-flavonoids (stilbenes and phenolic acids). Flavonoids have two rings of six carbon atoms, whereas non-flavonoids have a single ring of six carbon atoms [8]. In particular, flavonoids have been recognized as powerful antioxidants acting as scavenging free radicals and reducing oxidative reactions [9]. For that reason, these compounds have been associated with beneficial effects in humans, especially in the treatment and prevention of several chronic diseases, including cancer and inflammatory events [10]. Besides, grape marc is rich in carbohydrates, including monosaccharides (glucose and fructose) and structurally complex polysaccharides (pectins, heteroxylans, xyloglucan, and cellulose) [11], which can be used as a source for the production of ethanol and biogas [1]. Additionally, unsaturated fatty acids, mainly linoleic and oleic acid, have been identified in grape marc [12]. Due to its high content of phytochemicals, several researches have focused on reusing and applying grape marc in a wide range of sectors such as cosmetics, food, and pharmaceuticals [13]. In general, less than one-third of grape marc is destined to the distilleries to obtain alcoholic beverages (i.e., grappa) and a little part is sold to the companies that produce ethanol, tartaric acid and grape seed oil. Alternative uses of grape marc involve the agronomic exploitation as fertilizer or its use as biomass for energetic purposes but this material is still underexploited [14].

Nowadays, emerging technologies are attracting increasing interest by several food industries due to their potential to recover effectively and sustainable high-value-added compounds from grape marc with less consumed energy, thus replacing conventional heat-based processing methods [15]. The reuse of these wastes implies two operative steps: extraction from solid waste and the fractionation of the target compounds. In the case of extraction, the conventional technique is the extraction by Soxhlet or the use of maceration [16]. On the other hand, several non-conventional technologies to improve the extraction processes of valuable compounds from winery wastes and by-products have been reported in the literature, including pulsed electric fields (PEF) [17], high voltage electrical discharges (HVED) [18], pulsed ohmic heating (POH) [19], microwave-assisted extraction (MAE) [20], subcritical fluid extraction (SbFE) [21], supercritical fluid extraction (SFE) [22], high-pressure processing (HP) [23] and ultrasound-assisted extraction (UAE) [24]. Recent trends toward novel alternatives for winemaking industry waste utilization, including novel methods of extraction and the utilization of the wastes or their compounds, have been recently reviewed by Portilla Rivera et al. [25]. On the other hand, the fractionation and concentration of bioactive compounds is performed by conventional techniques, which include adsorption, chromatographic techniques, electrophoresis and vacuum distillation [15]. These techniques are characterized by high operating costs and energy consumption or the use of high temperatures that can degrade thermosensitive bioactive compounds. In this context, the use of pressure-driven membrane operations is a promising alternative for the purification and concentration of bioactive compounds present in wastewaters, waste extracts and other by-products generated in agro-industrial activities. The advantages of these processes include low energy and no-additives requirements, mild temperature and pressure conditions, continuous operation without changing the solvent phase, high separation efficiency and easy scale-up [26]. In particular, the use of nanofiltration (NF) membranes has been largely investigated for the fractionation and concentration of flavonoids, oligosaccharides, anthocyanins, carotenoids and phenolic compounds at low temperatures from vegetable extracts including apple pomace [27], pequi (*Caryocar brasiliense* Camb.) [28], propolis [29], graviola (*Annona muricata* L.) [30], jussara (*Euterpe edulis*) [31], artichoke [32], wine lees [33,34] and grape pomace [35,36,37,38,39] extracts.

In this context, this work was aimed at obtaining concentrated extracts enriched in bioactive compounds from Carménère grape marc through a combination of an extraction step assisted by ultrasound, a clarification of the extract by MF, and a concentration step by NF. In particular, the performance of three different NF membranes with a molecular weight cut-off (MWCO) in the range of 150 to 800 Da, was evaluated in terms of permeate flux and rejection towards polyphenols (in particular anthocyanins) and monosaccharides (glucose and fructose). The evaluation of membrane performance and the selection of adequate membrane pore size will allow designing future works oriented to the optimization of operating conditions or the inclusion of diafiltration to fractionate polyphenols and sugars from grape marc extracts.

## 2. Results and Discussion

### 2.1. Physico-Chemical Characteristics of Clarified Grape Marc Extract

The physico-chemical composition of the clarified grape marc extract is shown in Table 1. The total phenolics content in the clarified extract resulted of 469 mg gallic acid equivalent (GAE)/100 g dry weight, corresponding to about 1.46 g GAE/L. Similar values were obtained by Nayak et al. [40], which reported total phenolics content of 427.9 mg GAE/100 g for the water extraction of the Cabernet Sauvignon grape pomace (solid–solvent ratio of 1:20 *w*/*v*). Higher values were also reported by the same authors (801.6 mg GAE/100 g) by using a water–ethanol extraction (solid–solvent ratio of 1:20 *w*/*v*). Arboleda Meija et al. [37] have reported a total phenolics content of 260 mg GAE/100 g in the extract obtained from red grape pomace (a mixture of 60% Cabernet Sauvignon, 30% Sangiovese and 10% Syrah) through ultrasonic-assisted enzymatic extraction. Recoveries of about 1 g GAE/L extract have been reported in the extraction of grape marc Merlot by mechanical stirring with ethanol/water/hydrochloric acid (50:49:1) for 1 h [38].

The content of malvidin 3-*O*-glucoside (Mv-3-glc), the major anthocyanin found in grapes, resulted higher than that reported by Vitor Pereira et al. [39] in the pressurized liquid extraction of fresh Syrah grape marc (17.9 ± 0.2 mg/L). Higher values were detected for malvidin 3-(coumaroyl)-glucoside (75.71 ± 2.18 mg M3GE/L). Anyway, differences in the extraction yield of polyphenolic compounds from grape pomaces can be attributed to several technological factors, including grape variety and soil management, winemaking conditions and extraction technologies.

### 2.2. Permeate Flux Evaluation

Figure 1 shows the volumetric flux of permeate (*J_p_*) as a function of filtration time during the NF process of the microfiltered extract with the investigated membranes in the selected operating conditions. A rapid flux decay was observed for all selected membranes in the first 100 min of operation; after that, the drop in permeate flux was less pronounced. Decay in permeate flux can be attributed to the combined effect of extract components adsorption and pore occlusion [31] and the effect of increased concentration of retained compounds. However, among the NF membranes evaluated in this study, the NFX membrane presented the smallest molecular weight cut-off (MWCO); thus, there must have been less pore occlusion, with a smaller flux reduction.

It can be appreciated that there is a significant difference between the permeate flux of the NFX membrane and the other two membranes studied (F-ratio: 67.82; *p*-value = 0.000); indeed, the NFX membrane, with the lowest MWCO (150–300 Da), showed the lowest permeate flux values. On the other hand, these differences are not appreciated between the NFW (300–500 Da) and NFG (600–800 Da) membranes (F-ratio: 0.07; *p*-value = 0.788). Despite the differences in their MWCO the permeate flux after 100 min of operation results of the same order (about 27 L/m^2^h).

The average permeate flux values obtained for the Carménère clarified extract with selected membranes resulted lower than those reported recently by Arboleda Mejia et al. [37] in the NF of red grape pomace (a mixture of 60% cabernet sauvignon, 30% Sangiovese, and 10% Syrah) with cellulose acetate membranes. Authors reported average permeate fluxes between 43.38 and 50.58 L/m^2^h measured at 20 bar and 25 °C for three different membranes. On the other hand, for all selected membranes the observed permeate flux values resulted much higher than those reported by Vitor Pereira et al. [39] in the NF of microfiltered grape marc extract (obtained by pressured liquid extraction) with NF membranes of different material (polyamide (PA) and polyethersulphone (PES)) and MWCO (from 200 to 1200 Da) at an operating pressure of 40 bar. Indeed, initial permeate fluxes ranged from 1.3 L/m^2^h (for a PA membrane of 200–400 Da) to 11.4 L/m^2^h (for a PES membrane of 1000–1200 Da). Zaklis and Paraskeva [38] reported permeate flux values of about 15 L/m^2^h when treating an ultrafiltered grape marc extract with a spiral-wound PA membrane having a MWCO of 470 Da at an operating pressure of 10 bar. The observed differences can be attributed to different factors including marc characteristics, extraction conditions as well as the different membrane material and operating conditions. Since component separation occurs not only by size exclusion but also by diffusion in NF processes, the interactions of the membrane material with the extract and its target compounds should also be considered in these processes.

### 2.3. Rejection of Polyphenols

Figure 2 shows the rejection of the NF membranes towards total polyphenols as a function of filtration time: for all selected membranes the rejection increases after 60 min of operation. The ANOVA analysis shows that there are significant differences in the rejection between these membranes (F-ratio: 4.96; *p*-value = 0.012); however, this significant difference (F-ratio: 0.65; *p*-value = 0.424) were not appreciated between the NFW (300–500 Da) and NFG (600–800 Da) membranes. As expected, the membrane with the highest rejection (96.02%) was the NFX membrane, which is the membrane with the lowest MWCO (150–300 Da), followed by NFG (92.16%) and NFW (90.95%) membranes. Similar rejection values for total phenols were reported in the treatment of ultrafiltered grape marc extract with a polyamide membrane having a MWCO of 470 Da [38]. Díaz-Reinoso et al. [41] also found high rejection coefficients for phenolic compounds (95.3%) in the NF treatment of white vinasses. A strict correlation between the MWCO of PA NF membranes and the retention of total phenolic compounds has been recently reported by Yammine et al. [42] in the fractionation of polyphenols from grape pomace extracts. Basically, the rejection decreased from 100% up to 43% by increasing the MWCO range from 125–200 to 1000 Da. On the other hand, fluoropolymer membranes showed low average rejections towards phenolic compounds in comparison to PA membranes due to their higher hydrophobicity and lower fouling index.

In a previous work [43], 15 phenolic compounds were identified in the UAE extract from Carménère grape marc by HPLC-DAD-ESI-MS/MS: the majority of the compounds were anthocyanins, whereas two of them were identified as flavonols. Among them malvidin 3-(coumaroyl)-glucoside, malvidin-3-glucoside and malvidin 3-(acetyl)-glucoside were the most representative compounds. Table 2 shows the concentration of these compounds in the feed and permeate samples for the investigated membranes. All selected membranes showed high rejection towards these anthocyanins with values higher than 99.41%. In this regard, it should be pointed out that the total phenol analysis showed lower rejection because this analysis implies a colorimetric assay where several species can react, including not only anthocyanins compounds.

Yammine et al. [42] also reported high average rejections towards anthocyanins (95.9%) in the fractionation of polyphenols from grape pomace extracts with PA membranes in the range of 150–400 Da MWCO. Similarly, PA thin-film composite membranes with MWCO of 200–400 Da (NF200 and NF270 from Dow/Filmtec) exhibited anthocyanins rejections higher than 95% in the treatment of roselle extract within a range of operating pressure between 5 and 30 bar [44]. The NF270 membrane showed also a full rejection to anthocyanins in the treatment of microfiltered extracts of wine lees generated in the second racking of red winemaking [45] and more than 94% of rejection in the treatment of orange press liquor [46].

### 2.4. Rejection of Sugars

Table 3 shows the content of glucose and fructose measured in feed and permeate samples of NF tests performed with the investigated membranes and related rejections towards these compounds. The NFX membrane showed significant differences (F-ratio: 561.34; *p*-value = 0.000) with respect to the other two membranes in the rejection of both sugars, whereas there was no significant difference between NFG and NFW (F-ratio: 7.24; *p*-value = 0.055) membranes. In general, all the investigated membranes showed high rejections for both sugars. As expected, the NFX membrane, with the lowest MWCO (150–300 Da), exhibited the highest rejection for glucose and fructose (100 and 92.60%, respectively). On the other hand, NFW and NFG membranes showed similar rejection values for these compounds (of the order of 80% and 72%, respectively). For all selected membranes the glucose rejection was higher than the fructose rejection. This behaviour can be explained assuming that, at equilibrium, around 25% of fructose is in its furanose form, which results in an average size smaller than glucose [47].

A full rejection towards glucose and fructose was also detected in the treatment of bergamot juice with a spiral-wound PA membrane having a MWCO of 150–300 Da [48]. Similarly, Arboleda Mejia et al. [37] reported very high rejections towards glucose and fructose for some lab-made cellulose acetate NF membranes used in the treatment of red grape pomace extract. Yammine et al. [42] reported a decreased sugar rejection for PA membranes of increasing MWCO; in particular, the measured rejections were of 99% and 69% for membranes of 125–200 and 200–400 Da, respectively. 

The retention level measured for the investigated membranes is relatively high if compared with the molecular weight of glucose (180.1559 g/mol) and fructose (180.16 g/mol). This behavior can be explained assuming that other phenomena, other than the steric hindrance, affect the retention coefficients including interactions between the solutes and the membrane material as well as association of sugars with phenolic compounds [44].

As general trend, the tested membranes showed high rejection towards both sugars and anthocyanins; the separation of these compounds can be addressed using a continuous or discontinuous diafiltration process in order to contribute to reduce fouling phenomena with a consequent change in the selectivity so to promote the diffusion of sugars through the membranes. Futures works will be oriented to address this drawback in order to fractionate polyphenols and sugars adequately for future applications of fractionated extracts as a food additives.

## 3. Materials and Methods

### 3.1. Clarified Grape Marc Extract

Grape marc from red grape (Carménère variety) was supplied by Concha y Toro winemaker (Pencahue, Chile) and stored at −80 °C until use. An ultrasound-assisted extraction (UAE) was carried out in an ultrasonic system (SFX550 Sonifier, Branson Ultrasonics Corporation, Danbury, CT, USA) in optimal operating conditions: grape marc, 23.85% *w*/*w*; ethanol, 40% *w*/*w*; water, 36.15% *w*/*w*; amplitude, 20%; temperature, 22 °C; operating time, 15 min. The experimental set-up for the extraction process was illustrated in our previous work [5]. The extracted solution was filtered on nylon cloth and stored at −5 °C until its clarification by microfiltration (MF). The clarification was carried out using a mono-tubular ceramic membrane (Tami Industries, Nyons, France) with a pore size of 0.14 μm and an effective membrane area of 0.005 m^2^. The original hydraulic permeability of this membrane was 0.57 L/m^2^bar. The clarification process was operated at a transmembrane pressure (TMP) of 2.25 bar, a temperature of 25 °C and an axial feed flow rate (Q_f_) of 4.93 L/min according to the batch concentration configuration (recycling the retentate stream in the feed reservoir and collecting the permeate stream separately). 

### 3.2. NF Set-Up and Procedure

Experimental runs were performed by using a laboratory NF plant equipped with a stainless steel cell able to accommodate membranes in flat-sheet configuration (Figure 3). A heat exchanger, placed into the feed tank, was used to keep the feed temperature constant. Experiments were performed according to the batch concentration configuration. Each experimental run was stopped after 3 h of operation. Permeate flux was gravimetrically measured at different time intervals (each 10 min). The operating conditions were set as follows: TMP, 25 bar; Q_f_, 245.5 L/min; T, 30 °C.

Three polyamide thin-film composite membranes with MWCO in the range of 150–800 Da, all supplied by Synder Filtration (Vacaville, CA, USA) were used in this study. Their characteristics are reported in Table 4.

### 3.3. Analytical Measurements

Samples of UAE extracts, microfiltered extract (feed NF), permeates and concentrates from NF experiments were analysed for their anthocyanins content, glucose, fructose and total phenolic compounds.

#### 3.3.1. Anthocyanins Analysis

The identification of anthocyanins in the clarified extract was performed by HPLC-DAD-ESI-MS/MS. The HPLC-DAD conditions were: C18 column (250 mm × 4.6 mm, 5 μm), oven temperature 40 °C, injection volume 50 μL, mobile phase flow 0.5 mL/min, mobile phase composition water/acetonitrile/formic acid (87:3:10% *v*/*v*/*v*) (solvent A), and water/acetonitrile/formic acid (40:50:10% *v*/*v*/*v*) (solvent B). The ESI-MS/MS parameters were: positive ionization mode, 200–1200 *m*/*z* range, 4000 V of ionization voltage, the capillary temperature at 450 °C, nebulizer gas 40 psi, and auxiliary gas 50 psi.

The analyses of the anthocyanins in all samples from NF experiments were performed with a PerkinElmer AltusTM A-30 ultra-performance liquid chromatography UPLC^®^ System (PerkinElmer Inc., Waltham, MA, USA) equipped with an A-30 Quaternary Solvent Delivery Module, A-30 Sampling Module, A-30 PDA detector, A-30 FL detector, and A-30 RI detector. The separation was carried out using a Waters ACCQ-TAGTM ULTRA C18 column (2.1 mm × 100 mm, 1.7 μm). The chromatographic conditions were carried out according to Ruiz et al. [49] with slight modifications. The HPLC gradient transformation to UPLC gradient was made using the Columns Calculator software (Waters Corporation, Milford, MA, USA). The mobile phase was constituted by water/acetonitrile/formic acid (87:3:10% *v*/*v*/*v*) (solvent A) and water/acetonitrile/formic acid (40:50:10% *v*/*v*/*v*) (solvent B). The flow rate was 0.5 mL min^−1^, the column temperature 40 °C, and the injection volume was 2.0 μL. The gradient program was as follows: from 6 to 30% of B in 2 min, from 30 to 40% of B in 3 min, up to 60% of B in 0.5 min, and followed by 2.5 min of stabilization at 6% of B. The total run was 8 min. The detection was at 520 nm, and for quantification, a malvidin-3-glucoside external calibration curve was used. Results were expressed as malvidin-3-glucoside equivalents.

#### 3.3.2. Glucose and Fructose Quantification

The quantitative determination of glucose and fructose was carried out by a UPLC^®^ system (PerkinElmer Altus TM A-30 (PerkinElmer Inc., Waltham, USA) equipped with a Brownlee Analytical Amino column (4.6 mm × 150 mm, 5 µm). The samples were eluted using a mixture of acetonitrile/water (75:25). Each test’s conditions were a flow rate, injection volume, and temperature of 1 mL/min, 5 μL, and 40 °C, respectively. An external calibration curve was used for the quantification of glucose and fructose.

#### 3.3.3. Analysis of Total Phenolic Compounds

Total phenolic compounds (TPC) were estimated colorimetrically by using the Folin-Ciocalteau method [50], which is based on reducing tungstate and/or molybdate in the Folin-Ciocalteau reagent by phenols in an alkaline medium resulting in a blue-colored solution. Results were expressed as mg gallic acid equivalents per g of sample dry weight (mg GAE/g). The absorbance measurements were performed by using a UV/VIS/NIR Lambda 750 spectrometer (PerkinElmer Inc., Waltham, MA, USA) at 765 nm.

#### 3.3.4. Analysis of Turbidity and Total Soluble Solids

Turbidity measurements were performed at 20 °C by using a turbidimeter (Hanna Instruments Inc., Smithfield, RI, USA). Results were expressed as Nephelometric Turbidity Unit (NTU). Total soluble solids were measured by using a hand refractometer (Atago Co., Ltd., Tokyo, Japan) with a scale range of 0–32 °Brix.

### 3.4. Performance Parameters

The volumetric flux of permeate was calculated using Equation (1):(1)$$J_{p}=\frac{V_{p}}{A\cdot t}$$
where *J_p_* is the volumetric flux of permeate (L/m^2^h); *V_p_* is the volume of collected permeate (L); *t* is the time (h), and *A_p_* is the membrane permeation area (m^2^).

The degree of retention of component i by the membrane, defined as rejection (*R*), was calculated according to Equation (2):(2)$$R=\left(1-\frac{C_{p}}{C_{f}}\right)\cdot 100$$
where *C_p_* is the concentration of solute i in the permeate (mg/L); *C_f_* is the concentration of solute i in the feed (mg/L).

### 3.5. Data Analysis

The physicochemical analysis of the clarified grape marc extract and related samples obtained in the NF tests were expressed as mean ± standard deviation (SD) of three replicates. The analysis of differences between the tested membranes in terms of flux and rejection of target compounds was performed using ANOVA analysis at a 95% confidence level (α = 0.05). Statgraphics Centurion XVI (Statgraphics Technologies, The Plains, VA, USA) was used for all the computations.

## 4. Conclusions

Three commercial NF membranes, with MWCOs ranging from 150 to 800 Da, were tested for the recovery of anthocyanins and monosaccharides from clarified extracts of Carménère grape marc. Membranes showed differences in their performance in terms of permeate flux and rejection of target compounds. Among the investigated membranes, the NFX membrane, with the lowest MWCO, showed the lowest productivity (~15 L/m2h as steady-state permeate flux value) in the selected operating conditions. Permeate fluxes measured for NFG and NFW membranes resulted in two-fold higher than those measured for the NFX membrane. All membranes showed rejection values higher than 99.41% for the most representative anthocyanins detected in the Carménère grape marc extract (malvidin-3-O-glucoside, malvidin 3-(acetyl)-glucoside and malvidin 3-(coumaroyl)-glucoside). The studied membranes have also shown high rejections for glucose and fructose. In particular, the NFX membrane showed the highest rejection values (100% and 92.60% for glucose and fructose, respectively), whereas the NFW membrane was the one with the lowest rejection (80.57% and 71.62%, respectively). Despite the fact that this membrane seems to be more suitable for the separation of anthocyanins from sugars (higher rejection towards anthocyanins and lower rejection towards sugars) as a general trend, the tested membranes did not show a preferential rejection of anthocyanins over sugars. Indeed, the similar molecular weights of target compounds present in the clarified extract hinder a suitable membrane fractionation. Therefore, all tested membranes were suitable for concentration purposes.

Future studies will be oriented in the application of diafiltration and the optimization of the NF operating conditions to improve the separation capabilities of the membranes towards valuable compounds of grape marc extract. The combination of tight ultrafiltration membranes and the investigated membranes appears another useful strategy to reach these objectives.

## Figures and Tables

**Figure 1 molecules-26-02003-f001:**
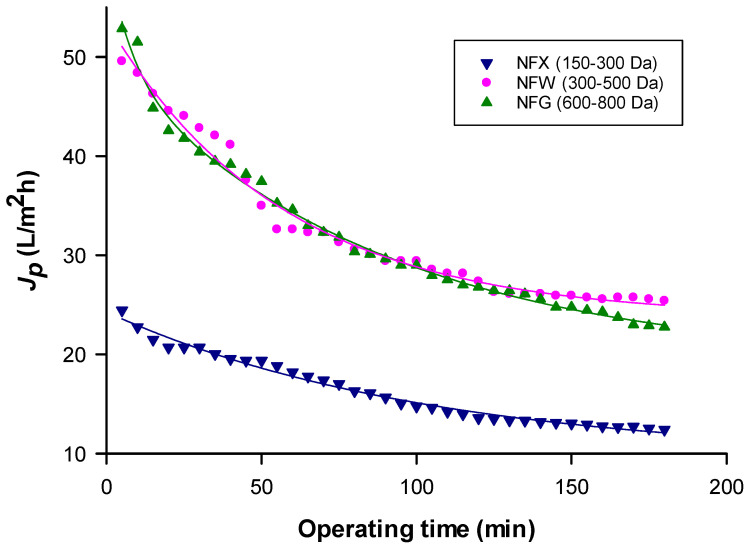
Time course of permeate flux (*J_p_*) for the investigated membranes. Operating conditions: Transmembrane pressure (TMP), 25 bar; Flow rate (Q_f_), 245.5 L/min; Temperature (T), 30 °C.

**Figure 2 molecules-26-02003-f002:**
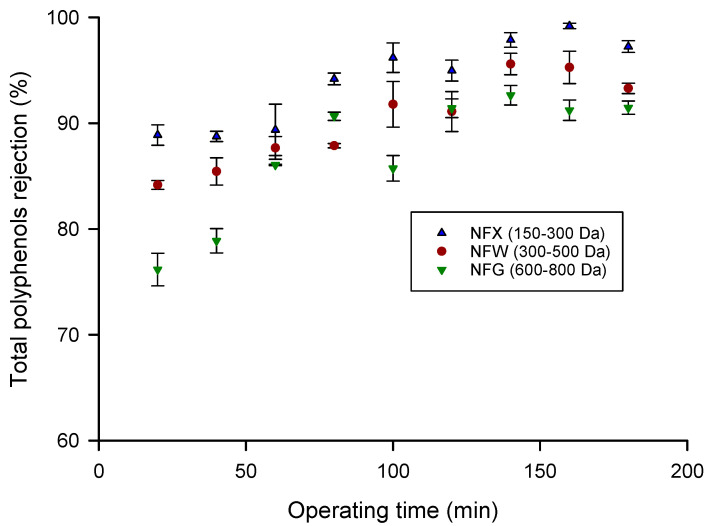
Time course of polyphenols rejection for selected membranes.

**Figure 3 molecules-26-02003-f003:**
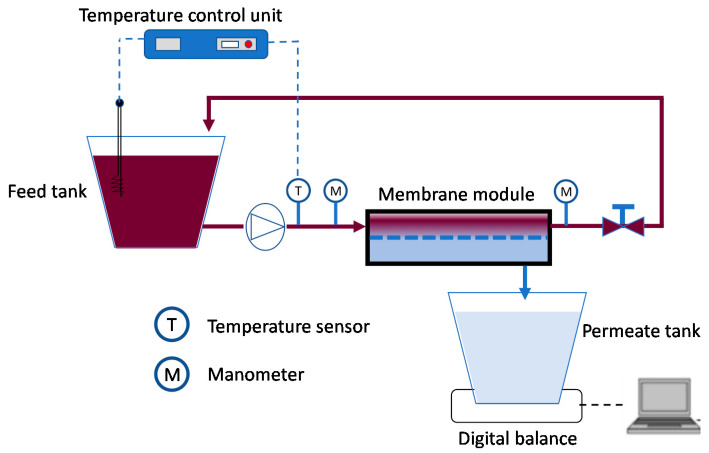
Scheme of the NF experimental set-up.

**Table 1 molecules-26-02003-t001:** Physico-chemical composition (mean ± standard deviation) of grape marc extract (M3GE, malvidin-3-glucoside equivalents; GAE, gallic acid equivalents).

Parameter	Value
Malvidin-3-*O*-glucoside (mg/L)	42.95 ± 0.71
Malvidin 3-(acetyl)-glucoside (mg M3GE/L)	18.01 ± 0.39
Malvidin 3-(coumaroyl)-glucoside (mg M3GE/L)	75.71 ± 2.18
Glucose (mg/L)	502.16 ± 19.68
Fructose (mg/L)	567.36 ± 25.69
Turbidity (Nephelometric Turbidity Unit, NTU)	5.58 ± 3.19
Total phenolic compounds (mg GAE/100g dry weight)	469 ± 17.00
Total soluble solids (°Brix)	17.60 ± 0.85

**Table 2 molecules-26-02003-t002:** Anthocyanins content in the feed and permeate stream of investigated NF membranes (M3GE, malvidin-3-glucoside equivalents) after 180 min of process.

Compounds	Sample	Membranes		
		NFX (150–300 Da)	NFW (300–500 Da)	NFG (600–800 Da)
Malvidin-3-*O*-glucoside (mg/L)	Feed	49.42 ± 0.35	44.43 ± 0.71	42.94 ± 1.24
	Permeate *	0.23 ± 0.09	n.d.	0.25 ± 0.10
	Retention (%)	99.53	100	99.42
Malvidin 3-(acetyl)-glucoside (mg M3GE/L)	Feed	21.51 ± 0.09	19.35 ± 0.42	18.29 ± 0.05
	Permeate *	0.09 ± 0.07	n.d.	0.08 ± 0.06
	Retention (%)	99.58	100	99.56
Malvidin 3-(coumaroyl)-glucoside (mg M3GE/L)	Feed	80.70 ± 0.15	73.92 ± 0.09	68.94 ± 0.42
	Permeate *	0.46 ± 0.29	n.d.	0.24 ± 0.29
	Retention (%)	99.43	100	99.65

* The value reported is the mean ± SD for the samples taken during 180 min of filtration (9 samples taken every 20 min).

**Table 3 molecules-26-02003-t003:** Glucose and fructose content in the feed and permeate stream for investigated NF membranes after 180 min of operation.

Compounds	Sample	Membranes		
		NFX (150–300 Da)	NFW (300–500 Da)	NFG (600–800 Da)
Glucose (mg/L)	Feed	516.34 ± 14.34	502.26 ± 19.68	502.26 ± 19.68
	Permeate	n.d	97.57 ± 38.07	97.24 ± 38.32
	Retention (%)	100	80.57	80.64
Fructose (mg/L)	Feed	494.72 ± 28.61	567.36 ± 25.69	567.36 ± 25.27
	Permeate	36.59 ± 0.45	161.04 ± 11.23	148.70 ± 11.88
	Retention (%)	92.60	71.62	73.79

**Table 4 molecules-26-02003-t004:** Characteristics of the selected NF membranes.

Membrane Type	Material	MWCO (Da)	Lactose Rejection (%) ^1^	MgSO_4_ Rejection (%) ^2^	NaCl Rejection (%) ^3^
NFX	PA TFC	150–300	99.0	99.0	40.0
NFW	PA TFC	300–500	98.5	97.0	20.0
NFG	PA TFC	600–800	60.0	50.0	10.0

MWCO, molecular weight cut-off; PA TFC, polyamide thin film composite. ^1^ Test Conditions: 2% lactose solution at 7.6 bar and 25 °C. ^2^ Test Conditions: 2,000 ppm MgSO_4_ solution at 7.6 bar and 25 °C. ^3^ Test Conditions: 2,000 ppm NaCI Solution at 7.6 bar and 25 °C.

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
