# Peer review of "Recovery of Anthocyanins and Monosaccharides from Grape Marc Extract by Nanofiltration Membranes"

_molecules, 2021, doi:10.3390/molecules26072003_

Round 1
Reviewer 1 Report
The paper entitled" Recovery of anthocyanins and monosaccharides from a clarified grape marc extract by nanofiltration" presents the possibility to recover polyphenols, in particular anthocyanins, and sugars from grape by-products by the use of nanofiltration. The paper is well-written and the results are clearly described. Some points must be improved before the publication of the paper.
In their introduction, from line 65 the authors describe polyphenols and flavonoids as important molecules with various biological activities. It could be interesting that the authors clarify their definition of polyphenols, flavonoids, anthocyanins.
In the part "Results" the authors present in Table 1 the physico-chemical composition of the clarified grape marc extract. Glucose, fructose and 3 anthocyanins are presented from the total phenolics. Were there no other flavonoids?
In Table 3 the authors show an higher retention percentage for glucose compared to fructose. The authors propose an association of sugars with phenolic compounds. Can they explain a little more?
Author Response
We truly appreciate all the constructive comments and suggestions from the reviewer. We have adopted all the suggestions in our revised manuscript. The modifications in the manuscript text are in yellow (background color). The following are our point-to-point responses to the reviewer's comments (responses are in red).
Point 1: In their introduction, from line 65 the authors describe polyphenols and flavonoids as important molecules with various biological activities. It could be interesting that the authors clarify their definition of polyphenols, flavonoids, anthocyanins.
Response 1: According to that recommended by the reviewer, a sentence in the introduction section was added to clarify the definition of polyphenols, flavonoids, anthocyanins (from lines 65 to 69).
Point 2: In the part "Results" the authors present in Table 1 the physico-chemical composition of the clarified grape marc extract. Glucose, fructose and 3 anthocyanins are presented from the total phenolics. Were there no other flavonoids?
Response 2: Among the identified anthocyanins investigated (in total 15), we have reported in Table 1 the most representative ones.
Point 3: In Table 3 the authors show an higher retention percentage for glucose compared to fructose. The authors propose an association of sugars with phenolic compounds. Can they explain a little more?
Response 3: The differences showed in Table 3 are explained because, in equilibrium, 25% of fructose is in its furanose form, which results in an average size smaller than glucose. A sentence (with a new reference) has been added in section 2.4 in order to clarify this result.
Reviewer 2 Report
General Comments
The objective of the research paper entitled ‘Recovery of Anthocyanins and Monosaccharides from a Clarified Grape Marc Extract by Nanofiltration’ was to obtain concentrated extracts enriched in bioactive compounds from Carménère grape marc through a combination ultrasound-assisted extraction and nanofiltration membranes. The performance of three different NF membranes with a molecular weight cut-off (MWCO) in the range of 150 to 600 Da, was tested in terms of permeate flux and rejection towards polyphenols (anthocyanins) and monosaccharides (glucose and sucrose) recovery.
The work carried out in this manuscript is good, and well written. The methodology can be repeated, the findings are constructive, conclusion supports the results of the study. There are some comments which will improve the manuscript and will need to be addressed before it is accepted for publication in Molecule journal.
The abstract is fine and well written.
Introduction section is very lenthy, lot of sentences can be removed. Make it crisp and tight. No need to provide a lot of background literature.
60 References are too many for a research article, I believe if you shorten your introduction, the number of references will go down.
Line 52, The worldwide production of what?
Line 115, through a combination of UAE and NF…
Line 123, Results and Discussion,
Line 136-137, and of red cabbage by shaking with acidified water (H3PO4 0.05 M) in 12 h, followed by pre-filtration [53]… is not relevant here, comparing phenolic content of grapes (fruit) with vegetable? There are plenty of study on grapes. The sentence can be omitted.
Line 145, 502.16 ± 19.68 mg/L
Line 158, what are NFX, NFW, NFG? Please explain
Figure 1, the title on y axes should be full; what is Qf in the title?
Table 2, 19.35±0.420, delete extra digit in the SD figure, keep it two points after decimal
Line 270, 22°C…space between number and unit
Table 4, foot note, MgSO4, number should be in subscript
Line 303, 40°C.. …space between number and unit
Line 318, min–1, number should be in superscript
Author Response
We truly appreciate all the constructive comments and suggestions from the reviewer. We have adopted all the suggestions in our revised manuscript. The modifications in the manuscript text are in yellow (background color). The following are our point-to-point responses to the reviewer's comments (responses are in red).
Point 1: Introduction section is very lenthy, lot of sentences can be removed. Make it crisp and tight. No need to provide a lot of background literature.
Response 1: We have shortened the Introduction section and also removed some references.
Point 2: 60 References are too many for a research article, I believe if you shorten your introduction, the number of references will go down.
Response 2: As reported above, we have reduced the number of references (from 60 to 50).
Point 3:Line 52, The worldwide production of what?
Response 3: The sentence was corrected by adding "Wine" in order to clarify this point.
Point 4: Line 115, through a combination of UAE and NF…
Response 4: The sentence was modified to clarify the different steps and the study of NF membranes for the concentration of anthocyanins and monosaccharides.
Point 5:Line 123, Results and Discussion,
Response 5: It was modified as suggested by the reviewer.
Point 6: Line 136-137, and of red cabbage by shaking with acidified water (H3PO4 0.05 M) in 12 h, followed by pre-filtration [53]… is not relevant here, comparing phenolic content of grapes (fruit) with vegetable? There are plenty of study on grapes. The sentence can be omitted.
Response 6: We agree with the reviewer. Thus, this sentence and related reference were removed.
Point 7: Line 145, 502.16 ± 19.68 mg/L
Response 7: It was modified as suggested by the reviewer.
Point 8: Line 158, what are NFX, NFW, NFG? Please explain
Response 8: These are the names of the membranes according to the manufacturer. In table 4, it was clarified
Point 9: Figure 1, the title on y axes should be full; what is Qf in the title?
Response 9: We have defined all the abbreviated terms, including Jp, in the caption of Figure 1.
Point 10: Table 2, 19.35±0.420, delete extra digit in the SD figure, keep it two points after decimal
Response 10: It was corrected accordingly.
Point 11: Line 270, 22°C…space between number and unit
Response 11: It was corrected accordingly.
Point 12: Table 4, foot note, MgSO4, number should be in subscript
Response 12: It was corrected accordingly.
Point 13: Line 303, 40°C.. …space between number and unit
Response 13: It was corrected accordingly.
Point 14: Line 318, min–1, number should be in superscript
Response 14: It was corrected accordingly.
Reviewer 3 Report
Comments and suggestions
- I propose introducing the following additions to table 1: - specification of the test conditions
- adding a pH parameter to the characteristics.
- report the value of total phenolic compounds per 100 g that this
value is quoted in the text
- For clarity added explanation MWCO (the molecular weight cut off) on page 4/158.
- 1 is illegible with respect to the type of membranes used, it is possible to describe the course of variability using a polynomial.
- 2 is illegible, in this form it cannot be accepted, what is the message for the reader in relation to the main purpose of this article, from this point of view, the most important thing is to show the differences in the registration of phenol content between the applied membranes.
- I suggest supplementing the information in Table 2 with the total operation time, adding the Retention item at Feed, Permeate like in Table 3.
- Provide the operation time in Table 3.
- The data on the characteristics of the Feed presented in Tables 1, 2 and 3, which was used for the research, are not compatible. the material should be homogeneous in terms of physical and chemical properties.
- The characteristics of an ultrasonic system should be supplemented with the data regarding power input and sonication time, p. 7/270.
- I have not found a method for describing the measurement of turbidity and other parameters, please fill in. I recommend that the described methods refer to the source.
General comments
Methodically, the work was done correctly.
The results obtained were predictable.
To some extent, the work provides new information on the results obtained.
However, in the field of nanofiltration, new elements and more diverse methods should be introduced.
Author Response
We truly appreciate all the constructive comments and suggestions from the reviewer. We have adopted all the suggestions in our revised manuscript. The modifications in the manuscript text are in yellow (background color). The following are our point-to-point responses to the reviewer's comments (responses are in red).
Point 1: I propose introducing the following additions to table 1: - specification of the test conditions
Response 1: The test condition and the methodology used for the Physico-chemical analysis are placed in section 3.
Point 2: adding a pH parameter to the characteristics.
Response 2: Unfortunately, we did not measure the pH
Point 3: report the value of total phenolic compounds per 100 g that this value is quoted in the text
Response 3: It was corrected accordingly.
Point 4: For clarity added explanation MWCO (the molecular weight cut off) on page 4/158.
Response 4: The term MWCO has been specified.
Point 5: 1 is illegible with respect to the type of membranes used, it is possible to describe the course of variability using a polynomial.
Response 5: We have modified Figure 1. Experimental data have been fitted through an exponential decay (fitting was better than using a polynomial correlation). The R2 value related to these fitting was 0.9856 (NFX membrane), 0.9927 (NFG membrane), and 0.9839 (NFW membrane). The y-scale range has been reduced from 0-60 to 0-55 in order to compare the data better.
Point 6: 2 is illegible, in this form it cannot be accepted, what is the message for the reader in relation to the main purpose of this article, from this point of view, the most important thing is to show the differences in the registration of phenol content between the applied membranes.
Response 6: We have modified Figure 2. The new Figure has been created as a scatter plot instead of a bar plot. In the new Figure, the behavior of each membrane can be clearly distinguished from the other one.
Point 7: I suggest supplementing the information in Table 2 with the total operation time, adding the Retention item at Feed, Permeate like in Table 3.
Response 7: The operation time, as well as the percentage of rejection, was included in Table 2.
Point 8: Provide the operation time in Table 3.
Response 8: It was provided accordingly.
Point 9: The data on the characteristics of the Feed presented in Tables 1, 2 and 3, which was used for the research, are not compatible. the material should be homogeneous in terms of physical and chemical properties.
Response 9: The Physico-chemical composition presented (Tables 1, 2, and 3) for the feed solution (Tables 1, 2, and 3) does not significantly differ from the statistical point of view. The differences have a normal variability considering the theory of sampling for each experience and variabilities attributed to the equipment and operations (i.e. dilution of samples due to water residues in the NF plant after cleaning and washing).
Point 10: The characteristics of an ultrasonic system should be supplemented with the data regarding power input and sonication time, p. 7/270.
Response 10: The description of the operating conditions for the UAE is placed in section 3.1 (from line 289 to 295)
Point 11: I have not found a method for describing the measurement of turbidity and other parameters, please fill in. I recommend that the described methods refer to the source.
Response 11: A new subparagraph (3.3.4) to describe the methods used for turbidity and total soluble solids measurements have been included.
Reviewer 4 Report
Line 22-23
- The NFX membrane, with the lowest MWCO (150-300 Da), showed the lowest permeate flux (~15 L/m2h as steady-state value).
I think that it is not very important item to say in the abstract. The researches know than smaller MWCO generates less flow.
Line 24- 25
All the membranes showed rejection higher than 99.41% for the quantified anthocyanins.
And then say:
whereas the NFW membrane (MWCO 300-500 Da) was the one with the lowest rejection for these compounds (80.57 and 71.62%, respectively).
The two phrases seem to be incongruous
Line 116
In particular, the performance of three different NF membranes 116 with a molecular weight cut-off (MWCO) in the range of 150 to 600 Da
In the abstract the values of MWCO are different
Line 126
-Authors do not define GAE the first time they use the acronym
Line 124-154, the repit the values (they have put a table 1, and before they write the values than are written in the table. Review this part. They should highlight only what is really interesting. It is not necessary to repeat the data
They do not show any section on the type of membranes used, filtration methodology, sampling, etc.
¿why and how do they work at 30ºC?
They work at 25 bares, it is a very high pressure ?why?
Author Response
We truly appreciate all the constructive comments and suggestions from the reviewer. We have adopted all the suggestions in our revised manuscript. The modifications in the manuscript text are in yellow (background color). The following are our point-to-point responses to the reviewer's comments (responses are in red).
Point 1: Line 22-23 The NFX membrane, with the lowest MWCO (150-300 Da), showed the lowest permeate flux (~15 L/m2h as steady-state value). I think that it is not very important item to say in the abstract. The researches know than smaller MWCO generates less flow.
Response 1: We have modified the sentence accordingly.
Point 2: Line 24- 25. All the membranes showed rejection higher than 99.41% for the quantified anthocyanins. And then say: whereas the NFW membrane (MWCO 300-500 Da) was the one with the lowest rejection for these compounds (80.57 and 71.62%, respectively). The two phrases seem to be incongruous
Response 2: The original phrase is, "All the membranes showed rejection higher than 99.42% for the quantified anthocyanins. Regarding sugars rejection, the NFX membrane showed the highest rejection for glucose and fructose (100 and 92.60%, respectively), whereas the NFW membrane (MWCO 300-500 Da) was the one with the lowest rejection for these compounds (80.57 and 71.62%, respectively)". Therefore, the lower rejections are referred to the sugars (glucose and fructose) and not anthocyanins.
Point 3: Line 116. In particular, the performance of three different NF membranes 116 with a molecular weight cut-off (MWCO) in the range of 150 to 600 Da. In the abstract the values of MWCO are different
Response 3: It was corrected accordingly.
Point 4: Line 126. Authors do not define GAE the first time they use the acronym
Response 4: The definition of the acronym has been included.
Point 5: Line 124-154, the repit the values (they have put a table 1, and before they write the values than are written in the table. Review this part. They should highlight only what is really interesting. It is not necessary to repeat the data
Response 5: We have modified the text removing data reported in Table 1. We have reported the data of phenolic compounds in the text expressed in mg GAE/100 g and g GAE/L to better compare our data with data reported in the literature.
Point 6: They do not show any section on the type of membranes used, filtration methodology, sampling, etc.
Response 6: The membranes' characteristics are placed in section 3.2 (Table 4), including procedures and methodologies of analytical measurements.
Point 7: why and how do they work at 30ºC?
Response 7: We assume that this temperature could be the optimum value to improve permeation flux and avoid thermal degradation of target compounds
Point 8: They work at 25 bares, it is a very high pressure ?why?
Response 8: We have not evaluated the effect of pressure on the performance of the investigated membranes. In any case, the operating pressure used was lower than the maximum pressure allowed for these membranes (41 bar at temperatures lower than 35 °C).
Round 2
Reviewer 4 Report
I suggest put the part of materials and methos before results.